# Knowledge and Poor Understanding Factors of Stroke and Heart Attack Symptoms

**DOI:** 10.3390/ijerph16193665

**Published:** 2019-09-29

**Authors:** Chang Hoon Han, Hyeyun Kim, Sujin Lee, Jae Ho Chung

**Affiliations:** 1Department of Internal Medicine, National Health Insurance Service Ilsan Hospital, Goyang 10444, Korea; hchoon7001@gmail.com; 2Department of Neurology, Catholic Kwandong University College of Medicine, International St. Mary’s Hospital, Incheon 22711, Korea; imkhy77@gmail.com (H.K.); jiniyamyam@naver.com (S.L.); 3Department of Internal Medicine, Catholic Kwandong University College of Medicine, International St. Mary’s Hospital, Incheon 22711, Korea

**Keywords:** cardiovascular disease, awareness, stroke, heart attack

## Abstract

Objectives: Adequate awareness of cardiovascular disease (CVD) may help in its prevention and control. Therefore, we evaluated knowledge among the general population of stroke and heart attack symptoms and determined the factors associated with poor understanding of CVD. Methods: This cross-sectional study included 228,240 adults (102,408 males, 125,832 females) who participated in the 2017 Korean Community Health Survey. Data on sociodemographic characteristics and cognizance of the warning signs of CVD events (stroke and heart attack) were examined. Logistic regression analysis was used to investigate factors associated with poor understanding of CVD. Results: The stroke and heart attack warning signs that were identified least often by respondents were “sudden poor vision in one or both eyes” (66.1%) and “pain or discomfort in the arm or shoulder” (53.8%). Of the subjects, 19.0% had low CVD knowledge scores (less than 4 out of 10) with males having lower scores than females. In the multivariate analysis, poor understanding of CVD warning signs was significantly associated with older age, male gender, lower education level, lack of regular exercise, unmarried status, unemployment, poor economic status, poor health behaviors (high salt diet, no health screening), poor psychological status (high stress, self-perceived poor health status), and the presence of hypertension or dyslipidemia. Conclusions: Specialized interventions, including those based on public education, should focus on groups with less knowledge of CVD.

## 1. Introduction

Cardiovascular diseases (CVD), such as stroke and heart attack, are the second most common cause of death in South Korea. In 2008, there were about 56.5 and 25.7 deaths per 100,000 South Koreans from stroke and heart attack, respectively [1]. Despite remarkable improvements in the treatment of heart attack and stroke, the burden of these two diseases remains high. Early recognition of the cardinal symptoms of these two diseases is important for appropriate management, to prevent poor clinical outcomes. Stroke and heart attack have common risk factors, such as smoking and alcohol consumption; it is important to assess the level of knowledge among the general population regarding the symptoms and signs of stroke and heart attack, to facilitate the early detection of CVD, because prompt treatment is associated with a good prognosis and good chance of survival after heart attack or stroke onset [2]. Greater knowledge of symptoms and signs of CVD was associated with more rapid access to emergency treatment [3], which may ultimately result in a better prognosis and increased survival rate. Therefore, to shorten the time from heart attack or stroke onset to hospital visit, it is important to improve the public’s knowledge of stroke warning symptoms and signs [4]. Identifying groups with poor understanding of CVD is the first step in targeted interventions aimed at reducing treatment delay. Some studies have reported on the awareness of the South Korean population regarding CVD, but these studies were either conducted in a limited area or examined a small population [5,6]. No nationwide study has been conducted on this issue in South Korea. Therefore, we determined the current level of knowledge of the signs and symptoms of heart attack and stroke among the general population of South Korea. We also investigated the factors associated with lower knowledge of these signs, using data from a nationwide, nationally representative population-based health survey, the 2017 Korean Community Health Survey (KCHS).

## 2. Methods

### 2.1. Study Participants

We obtained data from the 2017 KCHS (https://chs.cdc.go.kr/chs/index.do), which was conducted by the Korea Centers for Disease Control and Prevention (KCDC). The nationwide cross-sectional KCHS uses a two-stage sampling process. First, a probabilistic method is applied to select primary sampling units based on the number of households in various parts of South Korea. Then, households are systematically selected for evaluation in each sampling area. This two-stage sampling process guarantees representativeness of the entire South Korea population [7]. The KCHS survey data are weighted to ensure that all samples are statistically representative of the general population. All data were obtained by well-trained researchers and informed consent was obtained from all participants.

Of the total of 228,381 participants aged 19 to 110 years, we excluded 141 who did not complete the survey, i.e., for whom data were lacking on gender, age, smoking status, risky alcohol drinking, regular exercise, marital status, job, health insurance status, family income, region of residence, education level, body mass index (BMI), engagement in weight control management, saltiness of diet, self-perceived stress, self-perceived health status, health screening status, oral health screening status, or the presence of hypertension, diabetes mellitus, or dyslipidemia. Finally, 228,240 (102,408 males, 125,832 females) participants were included.

### 2.2. Socioeconomic and Demographic Factors

The socioeconomic and demographic factors included in the analyses were age; smoking status; risky alcohol drinking; regular exercise; marital status; job; health insurance status; family income; region; education level; BMI; engaged in weight control management; saltiness of diet; self-perceived stress; self-perceived health status; health screening status; oral health screening status; and the presence of hypertension, diabetes mellitus, and dyslipidemia. Age was categorized as ≤59 or ≥60 years old. Smoking status was categorized as non-smoker or smoker (past or current) [8]. Risky alcohol drinking was defined as consuming more than five alcoholic beverages on a single occasion more than 12 times during the last year [9]. Regular exercise was defined as walking at least five times per week, for at least 30 min each time (moderate exercise), or engaging in strenuous exercise at least three times per week for 20 min each time, as defined by the American College of Sports Medicine Guidelines [10]. Marital status was classified as married or other (single, divorced, separated, or widowed). Job status was categorized as employed or unemployed (including students and housewives). Health insurance status was categorized as national health insurance or medical aid/none. Family income was classified as below or above average. Residency was categorized as urban or rural. Education level was categorized into ≤ middle school or ≥ high school. BMI (kg/m^2^) status was categorized using the Asia-Pacific criteria (low BMI, <18.5 kg/m^2^; normal BMI, 18.5–24.9 kg/m^2^; and high BMI, ≥25 kg/m^2^) [11]. Behavior related to weigh control refers to both efforts to reduce weight and other weight management behaviors. A positive response to the question, “Have you tried to manage your weight in the last year?” was deemed indicative of weight management efforts. Self-perceived saltiness of diet was classified into five categories, very high salt, high salt, moderate, low salt, or very low salt, and subsequently re-categorized as high (very high, high), moderate, or low (low, very low) salt diet. Participants were asked the following question as a proxy for their stress levels: “Are you feeling stressed in your life?” Responses were categorized into five categories: no, some, moderate, high, or very high stress. The stress level was re-classified as low (no or mild stress) or high (moderate or severe stress) stress. Self-perceived health status was determined by the following question: “In general, how do you feel about your health?” and categorized as very good, good, fair, poor, or very poor. Health status was then re-classified as good (very good, good, fair) or bad (bad or very bad). Health screening in the last 2 years and oral health screening in the last year were investigated. The presence of comorbidities was determined by asking the respondents if they had ever been diagnosed with diabetes mellitus, hypertension, or dyslipidemia.

### 2.3. Assessment of Knowledge about CVD

Participants answered the following questions to determine their knowledge of stroke warning signs. “If you think that the following sentences are related to the symptoms of a stroke, answer ‘Yes’, or if you do not know, answer ‘No’. Answer ‘I do not know’ if you are not sure”. Warning signs included the following: (1) sudden numbness or weakness of the face, arms, or legs, especially on one side of the body; (2) sudden confusion or difficulty speaking or understanding speech; (3) sudden visual impairment in one or both eyes; (4) sudden difficulty walking, dizziness, or loss of balance or coordination; and (5) a sudden severe headache with no known cause. These five symptoms are well established in Korea for indexing stroke awareness [12]. To determine respondents’ knowledge of heart attack warning signs, they were asked the following question: “If you think that the following sentences are related to the symptoms of a heart attack, answer ‘Yes’, or if you do not know, answer ‘No’. Answer ‘I do not know’ if you are not sure”. Warning signs included the following: (1) pain in the neck, jaw or back; (2) feeling weak, light-headed or faint; (3) chest pain or discomfort; (4) pain or discomfort in the arm or shoulder; and (5) shortness of breath. One point was awarded for each positive response to the above statements. Scores on the individual items were summed. For the analysis, we calculated overall CVD warning signs knowledge scores for each participant, ranging from 0 to 5, for heart attacks and strokes. In previous studies, the proportions of participants with knowledge of at least two stroke warning symptoms and signs differed significantly [13,14]. Therefore, to enable meaningful comparison with these previous studies, a stroke knowledge cutoff score ≥3 was used to measure knowledge level in our study. A low score was considered as 0–2 points (<50% of maximum possible score) and a high score was 3–5 points (≥50%) [15]. Finally, the overall CVD warning signs knowledge score was calculated by summing the respondent’s scores for the heart attack and stroke warning signs components. The maximum score for both components combined was 10, and low, intermediate, and high scores were deemed as 0–4 (<50%), 5–7 (≤50% to <80%), and 8–10 (≤80% to 100%) points, respectively. Although this scale is arbitrary, it allowed knowledge levels to be meaningfully compared between groups. We evaluated the participants’ preferred treatment options for heart attacks and strokes with the following question, “What would you do if you thought you were having a heart attack or stroke?” The respondents chose one answer from among ‘call an ambulance’, ‘go to a hospital’, ‘go to an Oriental medicine hospital’, and ‘contact family’.

### 2.4. Data Analysis

To derive descriptive statistics, we summarized categorical variables as frequencies including proportions and continuous variables as means with standard deviations. The chi-square test and independent sample t-test were used for group comparisons. To identify factors that predicted poor understanding of CVD warning signs, multivariate logistic regression analysis was conducted. The odds ratios (ORs) and 95% confidence intervals (CIs) were obtained after adjusting for gender; age group; smoking status; regular exercise status; marital status; job; health insurance status; family income; region of residence; education level; BMI; weight control behaviors; saltiness of diet; self-perceived stress; self-perceived health status; health screening status; oral health screening status; and presence of hypertension, diabetes mellitus, and dyslipidemia. All statistical analyses were performed using SPSS for Windows software (ver. 21.0; SPSS Inc., Chicago, IL, USA). *p*-values < 0.05 were considered to indicate statistical significance.

## 3. Results

### 3.1. Socioedemographic Characteristics of the Participants

Of the 228,240 adults invited to take part in this study, 102,408 males and 125,832 females participated; Table 1 summarizes their sociodemographic characteristics. The overall mean age was 53.6 years and the females were significantly older. Smoking (current/former), risky alcohol drinking, regular exercise, and married status were more common in males. Females had higher education levels (*p* < 0.001). Economic status (living in an urban area, national health insurance, and above-average family income) was higher in males (*p* < 0.001). The mean BMI was 23.4 kg/m^2^, with men having higher values than women (24.0 vs. 22.8 kg/m^2^, *p* < 0.001). Females reported more good health behaviors (low-salt diet and engagement in weight control management) than males, although health screening, including for oral health, was more common in males. Psychological status (self-perceived high stress and self-perceived poor health) was worse in females (*p* < 0.001).

### 3.2. Knowledge of Stroke and Heart Attack Symptoms

Table 2 shows the knowledge of warning signs of heart attack and stroke according to sex. Regarding stroke, most respondents (80.4%) identified sudden confusion or trouble speaking or understanding others as a symptom of stroke, with females being more aware of this symptom than males (80.9% vs. 79.8%, *p* < 0.0001). A considerable proportion of respondents (33.5%) were unaware that sudden poor vision in one or both eyes cause can be a feature of stroke, with no sex difference seen in knowledge of this symptom. Fewer respondents were unaware of the other symptoms of stroke, such as sudden numbness or weakness in the face, arms, or legs (24.3%), sudden dizziness, difficulty walking, or loss of balance (22.7%), and sudden headache with no known cause (33.5%), with females having better knowledge than males.

Regarding symptoms of heart attack, most respondents (83.0%) identified chest pain or discomfort as a symptom of heart attack, with women being more aware of this symptom than men (82.8% vs. 83.2%, *p* = 0.014). Far fewer participants were unaware of other symptoms of heart attack, such as pain in the neck (26.6%), jaw or back (27.7%), feeling weak, light-headed or faint (31.8%), pain or discomfort in the arm or shoulder (46.2%), and shortness of breath (21.8%); nevertheless, for all of these symptoms, women had better knowledge than men.

Overall, knowledge of stroke and heart attack symptoms was relatively poor and suboptimal results, with mean scores of 3.7 and 3.5 out of 5 points, respectively, with women having better knowledge than men (*p* < 0.0001). The overall knowledge of CVD was also relatively poor and suboptimal results, with a mean score of 7.1 out of 10 points; again, women had better knowledge than men (*p* < 0.0001). The CVD knowledge score was low (≤4 points) in 19.0% of respondents with men having lower scores than women. Of all respondents, 79.4% and 83% would call an ambulance when someone had symptoms or signs of stroke and heart attack, respectively.

### 3.3. Factors Associated with Lower Knowledge of CVD Symptoms

In the multivariate logistic regression analysis (Table 3), older age (≥60 years) (OR 1.02, 95% CI 1.01–1.06), male (OR 1.20, 95% CI 1.16–1.24), risky alcohol drinking (OR 1.06, 95% CI 1.03–1.09), lack of regular exercise (OR 1.20, 95% CI 1.18–1.23), unmarried status (OR 1.52, 95% CI 1.49–1.56), unemployment (OR 1.15, 95% CI 1.12–1.18), medical aid or no national health insurance (OR 1.07, 95% CI 1.01–1.13), living in a rural area (OR 1.06, 95% CI 1.03–1.08), lower education level (OR 1.59, 95% CI 1.54–1.64), no weight control behaviors (OR 1.11, 95% CI 1.19–1.25), high-salt diet (OR 1.22, 95% CI 1.18–1.26), self-perceived high stress (OR 1.03, 95% CI 1.01–1.06), self-perceived poor health status (OR 1.10, 95% CI 1.07–1.13), no health screening (OR 1.33, 95% CI 1.30–1.37), no oral health screening (OR 1.1.4, 95% CI 1.11–1.17), and presence of diabetes (OR 0.95, 95% CI 0.91–0.98), hypertension (OR 1.05, 95% CI 1.03–1.09), and dyslipidemia (OR 1.24, 95% CI 1.20–1.28) were independently associated with a poor overall understanding of CVD.

## 4. Discussion

This large-scale, community-based study conducted in South Korea found that knowledge of CVD among the general population is suboptimal, with 19% of the respondents having a score indicating poor understanding. Older age, male gender, lower education level, lack of regular exercise, unmarried status, unemployment, poor economic status, poor health behaviors (high-salt diet, no health screening), poor psychological status (self-perceived high stress and self-perceived poor health), and presence of hypertension or dyslipidemia were independent predictors of lower CVD knowledge scores. There were many sex differences in the degree of knowledge of stroke and heart attack warning signs, and women typically had better knowledge of the warning signs of stroke and heart attack than men. These findings should inform policy decisions, especially in terms of education programs to combat CVD.

Other studies in other countries showed CVD knowledge as follows. African-American men who had lower levels of education and low income showed low CVD knowledge score [16]. Hawaiians in the United States reported moderate to high levels of heart attack and stroke symptom knowledge [17]. Cardiovascular risk factors and knowledge of symptoms among Vietnamese Americans showed that only 59% of Vietnamese Americans knew that chest pain was a symptom of heart attack [18].

Regarding heart attack, the most commonly recognized symptoms were chest pain and shortness of breath. The recognition rates for most other warning signs were lower than 70%. Approximately 50% of the participants did not know that pain or discomfort in the arms and shoulders was a heart attack symptom.

For stroke, more than 70% of the participants identified (1) sudden confusion, trouble speaking or understanding others, (2) sudden dizziness, difficulty walking or loss of balance trouble speaking, and (3) sudden numbness or weakness in face, arm or leg as stroke symptoms. This is consistent with a previous report [19].

The least identified stroke symptom was sudden blindness in one or both eyes, consistent with a study showing that impaired vision is the stroke warning symptom identified least often [20]. The multiple regression analysis showed that lower education level, being a smoker, and residing in a rural area were important predictors of poor understanding of CVD. As expected, participants with less education had less knowledge of CVD. This accords with previous observations [19] and can be explained by the fact that more literate groups have greater exposure to CVD health-related education programs and can thus more readily achieve an understanding of CVD. Current smokers showed a poor understanding of CVD, which is consistent with other studies [21,22]. Factors such as age and area of residence are related to poor understanding of CVD; this was also seen in our study. An association of old age with less knowledge of CVD has also been reported [23]. A significant association between the area of residence and knowledge of CVD was also reported: urban residents knew more about CVD than rural residents [24]. Knowledge is the first step toward early CVD intervention [25]. Other important factors include knowledge-based action; access to local community care; a good public transport; and rapid assessment, diagnosis, screening and treatment. The action phase following onset of CVD symptoms primarily refers to the pre-hospitalization period, which is reliant on the actions of the affected individuals themselves or bystanders. In agreement with another study [26], we found that most people with heart attack or stroke symptoms will call an ambulance. However, about 20% of those with symptoms of heart attacks or strokes perform other actions. Public education is effective for improving recognition of CVD symptoms and signs. A four-month education program for the elderly (average age, 75 years) was effective in improving the ability to recognize CVD symptoms and signs [27,28]. Therefore, public health education programs should target individuals likely to have less knowledge of CVD.

There were some limitations to this study. First, the use of a cross-sectional design meant that causality could not be inferred. Moreover, several variables were self-report only, so recall bias (over or underestimation) may have been present. Although there is no guarantee that good knowledge of CVD improves cardiovascular outcomes, such knowledge is necessary for individuals to make informed decisions about their health. Strengths of our study included the use of a nationally representative sample of South Koreans, with adjustment for potentially confounding factors. The large sample size allowed for more precise statistical adjustments.

In conclusion, in spite of small differences in CVD score, we observed suboptimal knowledge regarding the warning signs of CVD events (heart attack and stroke) in the general South Korean population. Older age, male gender, lower education level, lack of regular exercise, unmarried status, unemployment, poor economic status, poor health behaviors (high-salt diet, no health screening), poor psychological status (self-perceived high stress and self-perceived poor health), and the presence of hypertension or dyslipidemia independently predicted poorer understanding of CVD. Moreover, our study findings showed rural dwellers disadvantage in CVD knowledge scores when compared with urban dwellers, which is consistent with previous study [29].

Our study shows that there is a need to improve knowledge of CVD among the general South Korean population. Public health education needs to emphasize the less CVD knowledge. This study could serve as a reference regarding knowledge of CVD among South Koreans, to inform future public health campaigns. Public health education programs should target specific groups, such as the elderly (aged ≥60 years), who are at high risk of both heart attack and stroke.

## Figures and Tables

**Table 1 ijerph-16-03665-t001:** Sociodemographic and clinical characteristics of the study sample.

	Total (*n* = 228,240)	Male (*n* = 102,408)	Female (*n* = 125,832)	*p*-Value
Age (years)	53.6 ± 17.7	52.7 ± 17.1	54.4 ± 17.7	<0.001
≤59	138,690 (60.8)	64,588 (63.1)	74,202 (59.0)	
≥60	87,450 (39.2)	37,820 (36.9)	51,630 (41.0)	
Smoking status				<0.001
Smoker	81,992 (35.9)	75,388 (73.6)	6604 (5.2)	
Non-smoker	146,248 (64.1)	27,020 (26.4)	119,228 (94.8)	
Risky alcohol drinking				<0.001
Yes	50,307 (22.0)	37,875 (37.0)	12,432 (9.9)	
No	177,933 (78.0)	64,533 (63.0)	113,400 (90.1)	
Regular exercise				<0.001
Yes	116,697 (51.1)	54,904 (53.6)	61,793 (49.1)	
No	111,543 (48.9)	47,504 (46.4)	64,039 (50.9)	
Marital status				<0.001
Married	153,861 (67.4)	74,338 (72.6)	79,523 (63.2)	
Other (single, divorced, separated, widowed)	74,379 (32.6)	28,070 (27.4)	46,309 (36.8)	
Job				<0.001
Yes	144,451 (63.3)	77,501 (75.7)	66,950 (53.2)	
No	83,789 (36.7)	24,907 (24.3)	58,882 (46.8)	
Health insurance				<0.001
National health insurance	220,708 (96.7)	99,389 (97.1)	121,319 (96.4)	
Medical aid or none	7532 (3.3)	3019 (2.9)	4513 (3.6)	
Family income				<0.001
Less than average	158,349 (69.4)	69,469 (67.8)	88,880 (70.6)	
More than average	69,891 (30.6)	32,939 (32.2)	36,952 (29.4)	
Residency				<0.001
Rural	100,116 (43.9)	44,604 (43.6)	55,512 (44.1)	
Urban	128,124 (56.1)	57,804 (56.4)	70,320 (55.9)	
Education				<0.001
≤Middle school	80,709 (35.4)	27,263 (26.6)	53,446 (42.5)	
≥High school	147,531 (64.6)	75,145 (73.4)	72,386 (57.5)	
Body Mass Index, kg/m^2^	23.4 ± 3.3	24.0 ± 3.1	22.8 ± 3.3	<0.001
<18.5	10,429 (4.6)	2619 (2.6)	7810 (6.2)	
≥18.5 and <24.9	157,366 (68.9)	65,706 (64.2)	91,660 (72.8)	
>25	60,445 (26.5)	34,083 (33.3)	26,362 (21.0)	
Weight control trial				<0.001
No	91,037 (39.9)	45,129 (44.1)	45,908 (36.5)	
Yes	137,203 (60.1)	57,279 (55.9)	79,924 (63.5)	
Salt diet				<0.001
Low	54,392 (23.8)	22,821 (22.3)	31,571 (25.1)	
Moderate	117,408 (51.4)	50,263 (49.1)	67,145 (53.4)	
High	56,440 (24.7)	29,324 (28.6)	27,116 (21.5)	
Perceived stress				<0.001
Low	173,863 (76.2)	78,883 (77.0)	94,980 (75.5)	
High	54,377 (23.8)	23,525 (23.0)	30,852 (24.5)	
Perceived health status				<0.001
Not bad	179,522 (78.7)	84,979 (83.0)	94,543 (75.1)	
Bad	48,718 (21.3)	17,429 (17.0)	31,289 (24.9)	
Health screening (recent 2 years)				<0.001
Yes	166,308 (72.9)	74,775 (73.0)	91,533 (72.7)	
No	61,932 (27.1)	27,633(27.0)	34,299 (27.3)	
Oral health screening (recent 1 year)				<0.001
Yes	83,261 (36.5)	38,369 (37.5)	44,892 (35.7)	
No	144,979 (63.5)	64,039 (62.5)	80,940 (64.3)	
Presence of risk factors				
Diabetes mellitus	25,190 (11.0)	12,289 (12.0)	12,901 (10.3)	<0.001
Hypertension	62,485 (27.4)	27,846 (27.2)	34,639 (27.5)	0.037
Dyslipidemia	39,766 (17.4)	16,051 (15.7)	23,715 (18.8)	<0.001

Values are presented as number (%) or mean ± SD. Regular exercise was defined as moderate or strenuous exercise performed on a regular basis (>30 min at a time five times per week for moderate exercise; >20 min at a time five times per week for strenuous exercise) or walking >30 min at a time more than five times per week.

**Table 2 ijerph-16-03665-t002:** Knowledge about symptoms of heart attack and stroke according to sex among adults.

	Total (*n* = 228,240)	Male (*n* = 102,408)	Female (*n* = 125,832)	*p*-Value
Symptoms of stroke				
Sudden numbness or weakness in face, arm or leg	172,762 (75.7)	75,996 (74.2)	96,766 (76.9)	<0.001
Sudden confusion, trouble speaking or understanding others	183,509 (80.4)	81,683 (79.8)	101,826 (80.9)	<0.001
Sudden poor vision in one or both eyes	150,826 (66.1)	67,769 (66.2)	83,057 (66.0)	0.396
Sudden dizziness, difficulty walking or loss of balance	176,337 (77.3)	78,278 (76.4)	98,059 (77.9)	<0.001
Sudden headache with no known cause	151,724 (66.5)	66208 (64.7)	85,516 (68.0)	<0.001
Symptoms of heart attack				
Pain in the neck, jaw or back	144,468 (63.3)	63,818 (62.3)	80,650 (64.1)	<0.001
Feeling weak, light-headed or faint	158,019 (69.2)	70308 (68.7)	67,711 (69.7)	<0.001
Chest pain or discomfort	189,439 (83.0)	85,218 (83.2)	104,221 (82.8)	0.014
Pain or discomfort in arm or shoulder	122,869 (53.8)	54,525 (53.2)	68,344 (54.3)	<0.001
Shortness of breath	178472 (78.2)	79,343 (77.5)	99,129 (78.8)	<0.001
Knowledge of stroke symptoms (total score = 5)	3.7 ± 1.7	3.6 ± 1.7	3.7 ± 1.7	<0.001
Low score 0–2 points	50,276 (22.0)	23,476 (22.9)	26,800 (21.3)	
High score 3–5 points	177,964 (78.0)	78,932 (77.1)	99,032 (78.2)	
Knowledge of heart attack symptoms (total score = 5)	3.5 ± 1/7	3.5 ± 1.7	3.5 ± 1.7	<0.001
Low score 0–2 points	58,375 (25.6)	26,892 (26.3)	31,483 (25.0)	
High score 3–5 points	169,865 (74.4)	75,516 (73.7)	94,349 (75.0)	
Overall cardiovascular disease score (total score = 10)	7.1 ± 3.2	7.1 ± 3.2	7.2 ± 3.2	<0.001
Low score 0–4 points	43,458 (19.0)	20,079 (19.6)	23,379 (18.6)	
Mid-range score 5–7 points	55,151 (24.2)	25,669 (25.1)	29,482 (23.4)	
High score 8–10 points	129,631 (56.8)	56,660 (55.3)	72,971 (58.0)	
Respondent’s reaction to stroke symptoms and Knowledge about treatment				<0.001
Call an ambulance	181,190 (79.4)	81,451 (79.5)	99,739 (79.3)	
Take them to a hospital	33,154 (14.5)	16,284 (15.9)	16,870 (13.4)	
Take them to an oriental medicine hospital	1905 (0.8)	569 (0.6)	1336 (1.1)	
Contact family	11,991 (5.3)	4104 (4.0)	7887 (6.3)	
Respondent’s reaction to heart attack symptoms and Knowledge about treatment				<0.001
Call an ambulance	189,351 (83.0)	84,876 (82.9)	104,475 (83.0)	
Take them to a hospital	28,383 (12.4)	13,807 (13.5)	14,576 (11.6)	
Take them to an oriental medicine hospital	440 (0.2)	148 (0.1)	292 (0.2)	
Contact family	10,066 (4.4)	3577 (3.5)	6489 (5.2)	

**Table 3 ijerph-16-03665-t003:** Factors associated with overall low knowledge for cardiovascular disease.

	OR (95% CI)
Age, years	
≤59	Reference
≥60	1.02 (1.01–1.06)
Gender	
Female	Reference
Male	1.20 (1.16–1.24)
Smoking	
No	Reference
Yes	1,02 (0.98–1.05)
Alcohol	
No	Reference
Yes	1.06 (1.03–1.09)
Regular exercise	
No	Reference
Yes	1.20 (1.18–1.23)
Marital status	
Married	Reference
Other (single, divorced, separated, widowed)	1.52 (1.49–1.56)
Job	
Yes	Reference
No	1.15 (1.12–1.18)
Health insurance	
National health insurance	Reference
Medical aid or none	1.07 (1.01–1.13)
Family income	
More than average	Reference
Less than average	1.03 (0.99–1.05)
Residency	
Urban	Reference
Rural	1.06 (1.03–1.08)
Education	
≥High school	Reference
≤Middle school	1.59 (1.54–1.64)
Body mass index, kg/m^2^	
≥18.5 and <24.9	Reference
<18.5	1.04 (0.90–1.10)
>25	0.94 (0.92–0.96)
Weight control trial	
Yes	Reference
No	1.11 (1.19–1.25)
Salt diet	
Low	Reference
Moderate	1.07 (1.05–1.10)
High	1.22 (1.18–1.26)
Perceived stress	
Low	Reference
High	1.03 (1.01–1.06)
Perceived health status	
Not bad	Reference
Bad	1.10 (1.07–1.13)
Health screening (recent 2 years)	
Yes	Reference
No	1.33 (1.30–1.37)
Oral health screening (recent 1 year)	
Yes	Reference
No	1.14 (1.11–1.17)
Presence of diabetes mellitus	
No	Reference
Yes	0.95 (0.91–0.98)
Presence of hypertension	
No	Reference
Yes	1.05 (1.03–1.09)
Presence of dyslipidemia	
No	Reference
Yes	1.24 (1.20–1.28)

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
