# Peer review of "Knowledge and Poor Understanding Factors of Stroke and Heart Attack Symptoms"

_ijerph, 2019, doi:10.3390/ijerph16193665_

Round 1

Reviewer 1 Report

General Comments

The research addresses an important population health issue. The research questions are well framed and literature-based in my opinion.  The Methodology is well described (see below for specific exception) and the conclusions are generally supported by the findings.

Specific Comments

Page 3 lines 32-34. The text a stroke knowledge cutoff score ≥ 2 is not consistent with your definitions of a low (0-2) and a high (3-5) score. Should this not read either a stroke knowledge cutoff score ≤ 2 or a stroke knowledge cutoff score of ≥ 3

Page 7 (erroneously labelled Page 1 in the electronically generated proof document), paragraph 1, lines 1-3. Given your definitions in the Methodology section of high (3-5) and low (0-2) scores, it is logically inconsistent to then describe scores of 3.7 and 3.5 as relatively poor. The same applies to your characterisation of a score of 7.1 for the aggregate CVD score. Better to use the term suboptimal and to reflect this in the discussion and conclusions

Suggested Recommendation

Also, as findings noted a rural dweller disadvantage in knowledge scores when compared with urban dwellers, it would enhance the paper's conclusions and usefulness for policy evidence if the authors also called for further research into the broader structural determinants of that disadvantage. See, for example, the general recommendations in the 2012 paper below:

Upstream, midstream, and downstream mechanisms in the creation of health inequalities should be extensively explored to indicate entry points of policies. Intervention programs intensive enough to significantly improve the health status of the disadvantaged should be newly developed, tested, and applied. 

J Korean Med Sci. 2012 May; 27(Suppl): S33–S40. Published online 2012 May 18. doi: 10.3346/jkms.2012.27.S.S33 PMCID: PMC3360172 PMID: 22661869

Health Inequalities Policy in Korea: Current Status and Future Challenges

Young-Ho Khang and Sang-il Lee

Author Response

Page 3 lines 32-34. The text a stroke knowledge cutoff score ≥ 2 is not consistent with your definitions of a low (0-2) and a high (3-5) score. Should this not read either a stroke knowledge cutoff score ≤ 2 or a stroke knowledge cutoff score of ≥ 3

Answer) We changed ≥ 3. Therefore, to enable meaningful comparison with these previous studies, a stroke knowledge cutoff score ≥ 3 was used to measure knowledge level in our study. A low score was considered as 0–2 points (< 50% of maximum possible score) and a high score was 3–5 points (≥ 50%)

Page 7 (erroneously labelled Page 1 in the electronically generated proof document), paragraph 1, lines 1-3. Given your definitions in the Methodology section of high (3-5) and low (0-2) scores, it is logically inconsistent to then describe scores of 3.7 and 3.5 as relatively poor. The same applies to your characterisation of a score of 7.1 for the aggregate CVD score. Better to use the term suboptimal and to reflect this in the discussion and conclusions. 

Answer) Thank you for sharp comments. As effect sizes (unlike p-values) are not directly affected by sample sizes, a statistically significant result can be obtained despite a small effect size as long as the sample is large enough. The superiority of large samples over small samples is probably one of the least controversial truths in research. However, as large enough sample sizes lead the results of studies based on large samples are often characterized by extreme statistical significance despite small or even trivial effect sizes. In such cases, statistical significance does not necessarily imply practical significance. With a very large sample, the standard error becomes extremely small, so that even minuscule distances between the estimate and the null hypothesis become statistically significant. As you commented, our study showed significant p-value inspite of small difference of CVD score. Our study has large participant may produce this result.

I corrected results part as follows

Overall, knowledge of stroke and heart attack symptoms was relatively poor and suboptimal results, with mean scores of 3.7 and 3.5 out of 5 points, respectively, with women having better knowledge than men (p < 0.0001). The overall knowledge of CVD was also relatively poor and suboptimal results, with a mean score of 7.1 out of 10 points; again, women had better knowledge than men (p < 0.0001)

I corrected conclusion part part as follows

In conclusion, in spite of small differences in CVD score, we observed suboptimal knowledge regarding the warning signs of CVD events (heart attack and stroke) in the general South Korean population.

Suggested Recommendation

Also, as findings noted a rural dweller disadvantage in knowledge scores when compared with urban dwellers, it would enhance the paper's conclusions and usefulness for policy evidence if the authors also called for further research into the broader structural determinants of that disadvantage. See, for example, the general recommendations in the 2012 paper below:

Upstream, midstream, and downstream mechanisms in the creation of health inequalities should be extensively explored to indicate entry points of policies. Intervention programs intensive enough to significantly improve the health status of the disadvantaged should be newly developed, tested, and applied. 

J Korean Med Sci. 2012 May; 27(Suppl): S33–S40. Published online 2012 May 18. doi: 10.3346/jkms.2012.27.S.S33 PMCID: PMC3360172 PMID: 22661869

Health Inequalities Policy in Korea: Current Status and Future Challenges

Answer) As your recommendation, I corrected as follows

Also, our study findings showed rural dwellers disadvantage in CVD knowledge scores when compared with urban dwellers compatible with previous study (26).

Reviewer 2 Report

The paper titled "Knowledge and poor understanding factors of stroke  and heart attack symptoms" is very important from medical and sociological point of view. 

In my opinion,  only minor revision is needed: Discussion should have more in formations  about similar studies from other countries. 

Author Response

The paper titled "Knowledge and poor understanding factors of stroke  and heart attack symptoms" is very important from medical and sociological point of view. 

In my opinion,  only minor revision is needed: Discussion should have more in formations  about similar studies from other countries.

Answer) I corrected as your recommendation as follows.

Other studies in other countries showed CVD knowledge as follows. African-American men who had lower levels of education and low income showed low CVD knowledge score (16). Hawaians in United State reported that moderate to high levels of heart attack and stroke symptoms knowledge (17). Cardiovascular risk factors and knowledge of symptoms among Vietnamese Americans showed that only 59% of Vietnamese Americans knew that chest pain was a symptom of heart attack (18).
